# Monitoring the Quality and Perception of Service in Colombian Public Service Companies with Twitter and Descriptive Temporal Analysis

**Dante Conti [1], Carlos Eduardo Gomez [2,*], Juan Guillermo Jaramillo [2,*] and Victoria Eugenia Ospina [2]**

[1] Departament of Statistics & Operations Research, Universidad Politécnica de Cataluña, 08034 Barcelona, Spain; dante.conti@upc.edu

[2] Maestría en Gestión de Información, Escuela Colombiana de Ingeniería Julio Garavito, Bogotá 111116, Colombia; victoria.ospina@escuelaing.edu.co

\* Correspondence: carlos.gomez-r@mail.escuelaing.edu.co (C.E.G.); juan.jaramillo-y@mail.escuelaing.edu.co (J.G.J.)

**Abstract:** The main goal of this research is to analyze the perception of service in public sector companies in the city of Bogota via Twitter and text mining to identify areas, problems, and topics aiming for quality service improvement. To achieve this objective, a structured method for data modeling is implemented based on the KDD methodology. Tweets from January to June 2022 related to the companies in the sector are processed, and a temporal analysis of the evolution of sentiment is performed based on the dictionaries Bing, AFINN, and NRC. Subsequently, the LDA algorithm (Latent Dirichlet Allocation algorithm) is used to visually identify the topics with the greatest negative impact reported by the users in each of the 6 months by adding the temporal dimension. The results revealed that, for Aqueduct (water supply service), the topic with the highest dissatisfaction is related to the "Water Tank Request" processes; for Enel (energy services) "Service Outages"; and for Vanti (gas services), "Case solution and request information". Temporal patterns of tweets, sentiments, and topics are also highlighted for the three companies.

**Keywords:** LDA; topic modeling; Twitter; temporal analysis; sentiment analysis

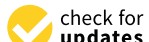



## 1. Introduction

In Bogota, the public service providers are Enel Colombia for power/electricity services [1], Vanti (gas supplies) [2], and Empresa de Acueducto de Bogota (water supply services) [3]. They are large companies with more than 2 million customers each, which should be characterized by timely service to their users, supported by customer-centric strategies.

However, according to the Superintendencia de Servicios Públicos de Colombia (Superintendency of Public Services of Colombia, SSPD) in its latest 2021 report [4], over 30% of the complaints received by this entity are attributed to those companies. Furthermore, it was found that during the past year, they were unable to respond to over 10,000 users, highlighting significant areas for improvement in the overall operational processes of the three corporations involved.

Under these premises, the main objective of this research is to identify the areas with the greatest opportunities for improvement that have a negative impact on the users' perception of service for public utilities in Bogota, based on the use of the VOC (Voice of the Customer) present in the clients' tweets, supported by the use of conventional text mining techniques and the temporal dimension that allow understanding the evolution of perception of service over time.

Twitter has positioned itself as one of the customer service channels with the highest growth for companies in the public services field in Bogota. This pattern has increased during and after the COVID-19 pandemic, as indicated by the sustainability reports. For

instance, Enel reported in 2020 a growth in publications, followers, and comments of more than 147% compared to 2019 [5].

Similar case studies demonstrate the feasibility and utility of this type of analysis for real-life applications in several fields. It is possible to find in the scientific literature some contributions that aim to achieve a similar goal as our study. A couple of examples can be listed. In [6], a study uses different classification methods (e.g., decision trees) to categorize customer comments based on their experience with tourism companies. In [7], research proposed how to detect problems and label opportunities for improvement processes. This was obtained from the comments of customers in the retail sector of some pharmaceutical companies on Twitter through the use of topic modeling, sentiment analysis, and visualization tools.

Regarding Bogota public service providers, there are no references focused on monitoring the quality and perception of service via Twitter. Our study intends to cover this gap, and at the same time, it incorporates descriptive temporal analysis to get a more dynamic overview related to the voice of the customers that interact with the three most important companies of public services in Bogota.

The potential of the voice of customers using Twitter as a digital customer service channel with its temporal patterns and evolution is the basis of this research, which integrates not only classical text mining and topic modeling but also the temporal dimension that enriches the results and facilitates their interpretability as support for intelligent decision-making.

The document is presented as follows: a first section where the main scientific background is exposed. This section is divided into subsections to better describe the state of the art according to the key concepts addressed in this research. The second section contains the material and methods. Then, the results of the temporal and trend analysis overview are presented. Additionally, the application of topic modeling (LDA), the critical topics, and their temporal evolution are determined. Finally, conclusions and future works are listed to close the article.

## 2. Background and Previous Research

### 2.1. Reference Frameworks or Similar Case Studies

During the last few years, interest in investigating how to exploit textual data has been increasing. This is due to the emergence of new sources of information such as social networks, where business users can express themselves and generate, for example, recommendations based on their experiences, as reported in [6].

One of the approaches related to leveraging the voice of the customer corresponds to the development of reference frameworks to improve understanding of the subject and the creation of baseline methodologies for processing comments or reviews by involving techniques such as data mining, text mining, or sentiment analysis, as presented in [8], where the research offers a focus on Twitter applications.

Some examples can be found in the scientific community. In [9], a model is presented containing four basic stages: data collection, management, classification, and visualization to lay a foundation in sentiment analysis and have certain basic elements of graphical representation on the perception of services in the public and private sectors in Rwanda. Furthermore, in [10], the article provides a comprehensive overview of the various technical approaches employed to classify tweets into positive, negative, or neutral sentiments. They not only describe each stage of the processing but also outline the overall benefits and drawbacks of this type of analysis. Additionally, that research meticulously outlines the methodology, providing a detailed account of each step.

In Colombia, there has been some research that uses Twitter as a data source to perform analysis; for example [11], proposes a neural network and machine learning model that personalizes the insurance service for each customer according to their needs. In this study, tools such as Tweet Archiver, OpenRefine, and Biome-textSE are applied, and metrics such

as precision, recall, and F1 are used to measure the performance of the model. Results are also deployed by using visualization tools such as dashboards.

## 2.2. Benchmarking with Text Mining

One of the applications of text mining and processing corresponds to the benchmarking or comparison of different attributes, products, or services of companies. For instance, the pioneering research proposed in [7] determines which areas are critical within companies in the UK retail sector. These areas were obtained by applying the LDA algorithm, sentiment analysis, and comparison of the main keywords of each topic with a theoretical review (performed manually) of the main factors of each area, which is similar to the CIER survey performed in Colombia [12].

On the other hand, ref. [13] details the use of a new method to analyze the perception of customers regarding the major features of the products of some cell phone brands. The methodology is described as MSAS—Microblog Sentiment Analysis System—and involves the definition of the main metrics to compare and process the tweets, the definition of the polarity, and the use of visualization tools to compare which attributes are more important.

Another piece of research that applies benchmarking in the field of satisfaction and service levels is explained in [14]. Here, a methodology is proposed to determine the level of detraction and potential promotion of three mobile network operators in South Africa. It highlights the use of different processing and analysis techniques, such as graphical methods to find trends, the definition of correlations based on timelines, and cluster definition of geo-referenced data. Additionally, topic modeling methods such as the LDA algorithm are applied to detect and classify the main topics obtained from mobile users.

One of the critical variables in the applications of text mining in benchmarks corresponds to the enrichment of the dictionaries to be used, as mentioned in [15]. This study proposes a model of predictability for followers of the Twitter pages of telecommunications companies in India after the entry of a new competitor to the market. When comparing the projections made with actual industry data, it is found that much of the success of the development is due to the enrichment of the dictionaries implemented.

In [16], lexicon enrichment in benchmarks also becomes highly relevant; an accuracy level of 73% is reached due to the implementation of a Naive Bayes-based model to identify improvement opportunities from customer feedback in the private transport sector.

## 2.3. Customer Perception by Using Text Mining

The use of text mining applied to customer perception has a great field of application because it helps measure the user's perception of a product, brand, or public figure, even though metrics or standard measurement scales have not been established in this field. A key study is presented in [17]. Here, an evaluation of the perception of Twitter users is proposed with the following metrics: user sentiment score, tweet sentiment score, positive-negative tweet ratio, and positive-negative user ratio. These are calculated from the classification of the tweets (positive and negative) of the followers of some public figures recognized worldwide through the LIWC tool (software for cognitive and emotional evaluation of texts) and the implementation of different dictionaries.

## 2.4. VOC and Text Mining

Sentiment analysis and the methods used to model and visualize textual data have been in constant evolution, and in some cases, new methods have been created or traditional methods have been combined to improve the accuracy and prediction of research results when quantifying user opinion.

This is reflected in the research of [8,18], respectively, where they apply a series of combinations of text classification algorithms (Naive Bayes, K Means, Fuzzy K Means, and Dirichlet Process) and data clustering (KNN, C4.5 Tree, and SMV) to find the best combination of methods and algorithms that best fits the texts to be analyzed. This is based on the best performance of various metrics that each model has and the application of

algorithms such as TD-IDF and LASSO to determine the main topics of customer reviews in fast food restaurants pre- and post-COVID-19 pandemic by adding the temporal dimension.

The combination of methods, techniques, and algorithms of text analytics provides tools for maximizing the use of VOC in social networks and indicates the basis and considerations of more in-depth evaluations. In [6], a proposal is based on the analysis of concordance between the comments of a hotel user and the rating provided by the customer in the surveys by implementing decision trees and classifiers such as Naive Bayes to predict the rating of a user from his review, obtaining an accuracy of 94.37%.

All references cited above are examples of the power of this type of analysis for real-life problems.

In Table 1, we provide an overview of the state-of-the-art that defines the scope of this research and its implicit theoretical fundamentals.

**Table 1.** Overview of research in the text mining field and similar.

| Application | Author | Field | Description |
|---|---|---|---|
| Reference framework | M. Ngaboyamahina and Y. Sun | Perception of public services in Rwanda | Definition of four general states in text analysis. |
| Reference framework | Kouloumpis, E., Wilson, T., and Moore, J | Sentiment analysis | The general process of applying sentiment analysis to a large corpus. |
| Reference framework | Avila Rodriguez, | Life insurance | Neural networks and machine learning models to understand and personalize life insurance tax. |
| VOCs in social media | Songpan | After-sales tourism | Comparison of hotel reviews and ratings using Naive Bayes and decision trees. |
| VOCs in social media | Bello-Orgaz, Menéndez, Okazaki, and Camacho | Furniture and decoration | Comparison of customer reviews using the C.4.5 Tree and clustering through the Dirichlet Process algorithm. |
| VOCs in social media | Kuo, Riantama, and Chen | Fast food | Main topics in customer reviews before and after the pandemic: comparison using TD-IDF and LASSO. |
| Customer perception | Ba and Lee | Perception of public figures | Perception of public figures, comparison metrics, and analysis over time. Use of statistical analysis through regression and comparison. |
| Benchmarks | Chamlertwat, Bhattarakosol, Rungkasiri, and Haruechaiyasak | Mobile phone | Technology Products: comparison of basic characteristics to know the customers' perception of the new products. |
| Benchmarks | Zhan, Han, Tse, Helmi Ali, and Hu | Pharmacy retail in the UK | Integration of customer pain points with operations based on sentiment analysis (Lexicon), topic modeling (LDA), and visual models (heatmap). |
| Benchmarks | Ranjan, Sood, and Verma | Telecomunications in India | Study of the impact on the followers of the pages of a new competitor in the market, applying correlation analysis and TD-IDF. |
| Benchmarks | Sari, Wierfi, and Setyanto | Public transport | Identification of the polarity of these on two Indian private transportation companies through online platforms using Naive Bayes, TF-IDF, and RapidMinker. |
| Benchmarks | Ogudo and Dahj Muwawa Jean | Mobile networks in South Africa | Determining and defining the level of potential detraction and promotion of three mobile network operators in South Africa via EDA and R. |

## 3. Materials and Methods

The KDD (Knowledge Discovery in Databases) methodology for data-driven modeling is used [19] in our research. This methodology considers an iterative and interactive process with databases or datasets. Different models or approaches are applied for data analysis to generate knowledge that can be used for decision-making or as input for new models or new analyses with new data [20]. This process is divided into 6 phases as follows: data and application domain, data extraction, data processing, data mining, results interpretation, and knowledge generation. In the following sections, the steps of the KDD will be explained in terms of this research. The first three steps belong to Section 3, and the final three are reported in Section 4.

### 3.1. Data and Application Domain

It is necessary to understand the progress and lessons learned of related works to data text mining on social media and VOC reported in Section 2. The purpose of this step is to define a process for data transformation that allows managing all variables involved to get results that could be reliable for supporting intelligent decision-making in the monitoring of perception and quality service for each of the three companies under study.

### 3.2. Data Extraction

The data for this research was extracted from the official Twitter accounts of the three companies. Table 2 summarizes the number of tweets extracted between 1 January and 30 June 2022. The sample downloaded corresponds to all tweets of the users who wrote to the official pages, which is considered concise with the scope of the research, i.e., a diagnosis-oriented frame of reference to apply solutions involving data mining in this sector.

**Table 2.** Number of tweets downloaded for each company.

| Public Basic Service | Company | Twitter Account | Number of Tweets |
| :---: | :---: | :---: | :---: |
| Electricity | Enel Colombia, Colombia, Bogotá. | @ codensaservicio @EnelColombia @EnelClientesCo | 8358 |
| Natural Gas | Vanti, Colombia, Bogotá. | @grupovanti | 2773 |
| Aqueduct | Acueducto y Alcantarillado, Colombia, Bogotá. | @AcueductoBogota | 12,479 |

### 3.3. Preparation and Data Mining

To process the textual data, the use of R software has been defined to obtain the DTM (Document Term Matrix) for each company. As part of the software definitions, a benchmark of R libraries or packages [21] for connecting and downloading the data using the Twitter API credentials was performed.

The dictionaries that will be used for the sentiment analysis (Table 3) are chosen, and then the method of application of each one is defined. A total of 1928 tweets were randomly chosen from the Enel [1] company to determine whether the method defined by R [21] or the assignment method (exhaustive term search) using the Merge function for each dictionary is better to obtain a greater coverage of the corpus. The results are presented below:

- AFINN Dictionary: Quantifies each word on a scale of −1 to 1 according to intensity;
- Bing Dictionary: Qualifies each word between positive, neutral, and negative;
- NRC Dictionary: Quantifies each word among one or more of the 8 sentiment categories on a scale of 1 to 10;
- Emoji Dictionary: Each emoji is assigned its equivalent in hexadecimal code to convert it into a word using a search.

**Table 3.** Results of the evaluation of methods for sentiment analysis.

| Dictionary | Syuzhet Library | | Merge (Assignment) | | Result |
|---|---|---|---|---|---|
| | No. Tweets Affected | % | No. Tweets Affected | % | |
| Bing | 948 | 49.17% | 1367 | 70.9% | Assignment |
| AFINN | 728 | 37.75% | 1226 | 63.59% | Assignment |
| NRC | 845 | 43.82% | 732 | 37.97% | RStudio |

Before data processing and before obtaining the DTM, 3 levels of cleaning are applied to better preprocess the data. These are the descriptions of the levels:

First level of cleaning (preprocessor): The variables of interest for the research are defined and the data structure for the modeling is configured. Additionally, the user threads are consolidated, the tweets from corporate, governmental, and non-profit organization accounts are eliminated, and finally, they are consolidated in a single database. On the other hand, emojis are transformed into words according to the hexadecimal code of the emoji and its equivalent in a single word. At this level, an average of 24.06% of the initially downloaded tweets are lost.

The second level of cleaning (the RemoveWords function): A list of terms identified in the messages that do not generate value for the research is defined, and using the "RemoveWords" function, they are eliminated from the messages. At this level, an average of 1.28% of the tweets from the first cleaning level are lost.

The third level of cleaning (StringR function): At this level, the StringR function is used, which is part of the process to obtain the DTM, and here the web addresses, hashtags, punctuation, accentuation, numbers, and StopWords defined within R [21] are eliminated. At this level, an average of 1.21% of the tweets resulting from the second level of cleaning are lost.

Considering the above points, a final DTM was obtained with the relevant information that was used to perform the models and analysis that will be described in the next section.

## 4. Results

### 4.1. Temporal Patterns

The collected data is used to perform a temporal analysis on the behavior of message traffic for each company via Twitter to get descriptive temporal patterns associated with the days of the week and the hours when users transmit the most quantity of messages (Figure 1).

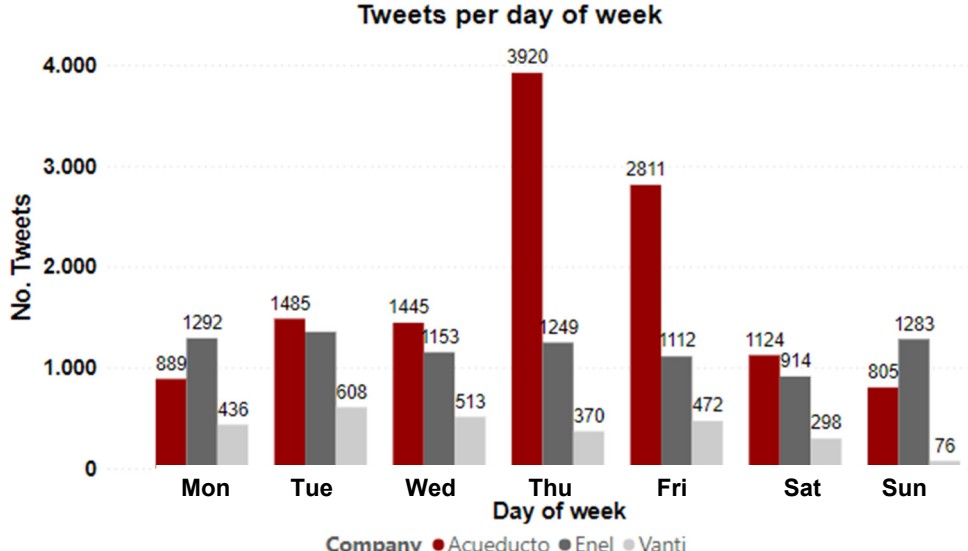

**Figure 1.** Number of tweets per day of the week for each public utility company.

In the case of the Acueducto, Thursdays and Fridays present the highest number of messages, with a marked trend for these days. On the other hand, Enel shows semi-homogeneous behavior on each of the days of the week. Finally, for Vanti, there is more traffic on Tuesdays and Wednesdays. It can be seen that day patterns are different within the three companies.

Regarding the hours of the day (Figure 2), the three companies have similar behavior related to the traffic of tweets. The highest traffic starts at 10 am and finishes around midnight. The lowest traffic is related to nighttime periods from midnight until 9 am, when there is a change in tendency.

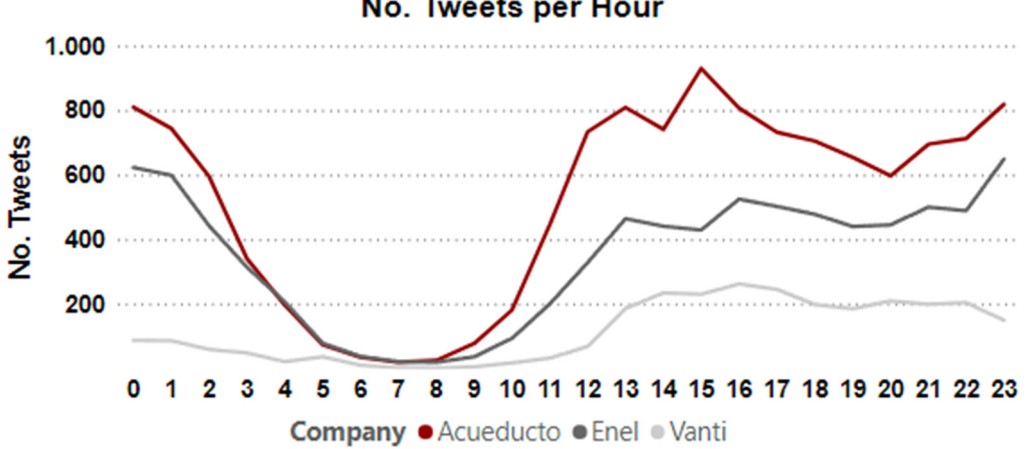

**Figure 2.** Tweet traffic per hour of the day for each company.

Trends, seasonality, and temporal patterns were also studied by plotting the number of tweets as a time series. Results for Enel are shown in Figure 3.

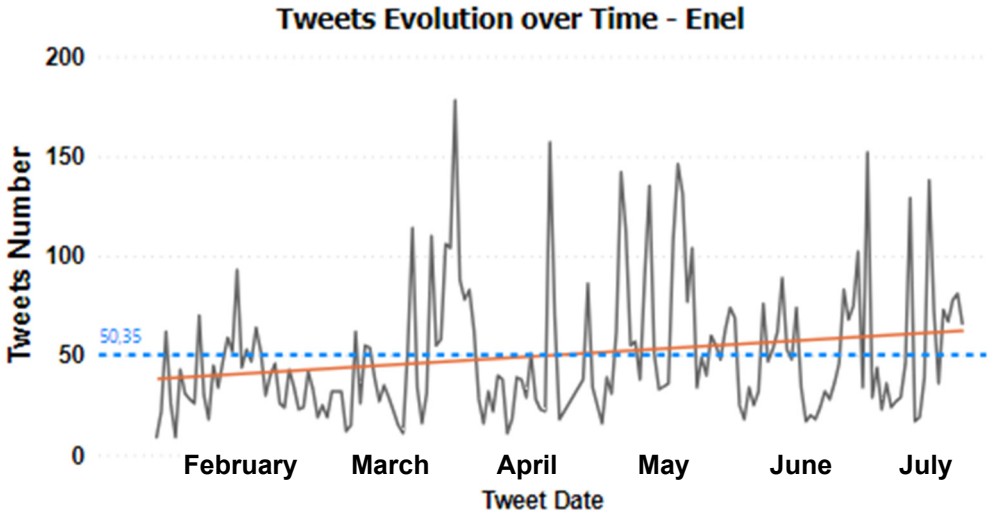

**Figure 3.** Enel time series with average (blue) and trend (orange).

In the case of Enel (Figure 3), an average of 50 daily tweets was obtained during the period analyzed, with an upward trend at the end of the period. When verifying the messages in the outlier points of the series, it can be observed that between the months of April and May, the daily number increases compared to the average since there is a rainy season in Bogota that causes power outages. Therefore, the rainy season is highly correlated with massive failures in the continuity of service in several locations of the city and the surrounding municipalities. In addition, people communicate the consequences of this service failure by talking about food conservation, the operation of vital equipment for people, and the damage to household appliances and electronic equipment.

For the Acueducto (Figure 4), at the end of the period analyzed, there were atypical values due to high contactability explained by a service outage in the northwestern area of Bogota, where users reported no service for more than 72 h, which was labeled as an outlier situation. For the rest of the studied period, the Acueducto time series seems to have homogenous behavior. However, the time series was reanalyzed by normalizing it. This process consisted of taking the seven values closest to the left of the first peak and the six values following the second peak to normalize the series and make the result consistent with the other data. The new time series is presented in Figure 5.

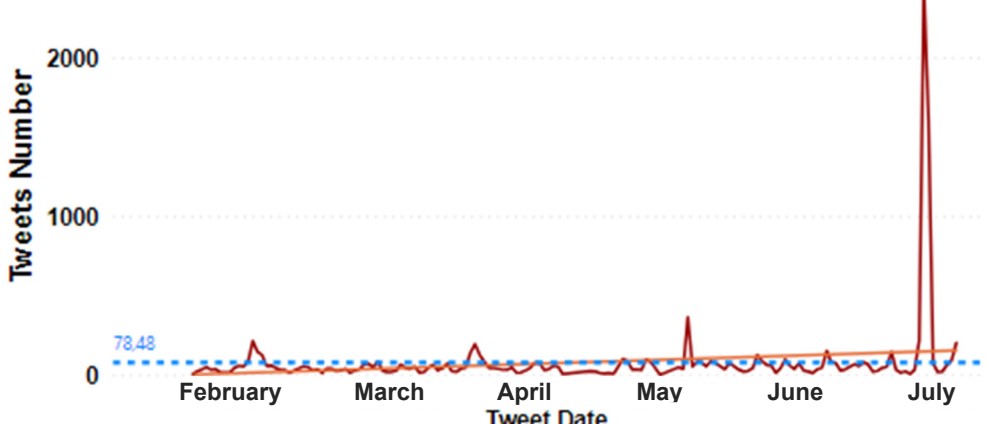

**Figure 4.** Acueducto time series.

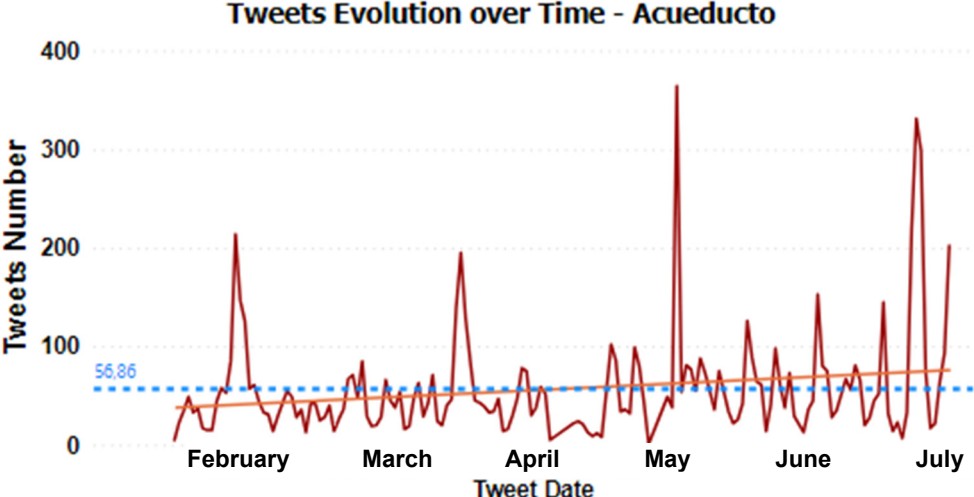

**Figure 5.** Normalized Acueducto with average (blue) and trend (orange).

From this analysis, Acueducto received 57 daily tweets on average with an upward trend at the end of the period, and a seasonality of the data is identified by observing that on average, every 47 days, the company shows high traffic of messages exceeding 200 tweets. The peaks identified in the series report problems related to short service outages, water quality problems, and mobility problems due to the execution of public networks by Acueducto.

In the case of Vanti, the results obtained in Figure 6 allow us to conclude that the time series shows an average of 18 daily tweets for the period analyzed, with a slight downward trend at the end of the period. With the volumetric data, it can be inferred that Twitter is not a representative channel compared to the other two companies. Atypical situations at the highest points were labeled when users reported reconnection and service costs,

home visits, and responses to customer requests through SICC (suggestions, issues, claims, or complaints).

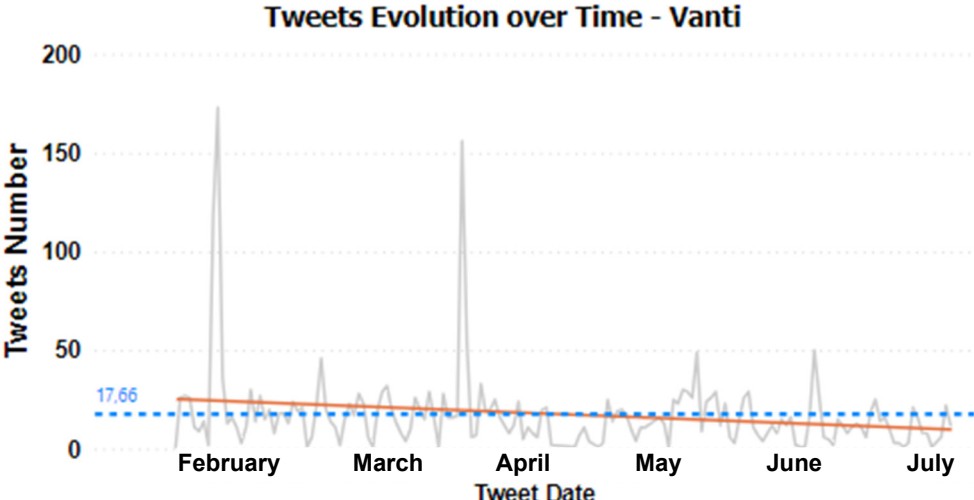

**Figure 6.** Vanti time series with average (blue) and trend (orange).

*4.2. Evolution and Trends of Sentiment Analysis*

4.2.1. Evolution of Sentiment with BING

The results of the labeling of the tweets in the consolidated corpus using the BING dictionary are shown in Table 4.

**Table 4.** Number of tweets categorized by company.

| Public Basic Service | Categorized Tweets | Unassigned Tweets | Total Tweets |
| --- | --- | --- | --- |
| Electricity | 10,802 (86.56%) | 1677 (13.43%) | 12,479 |
| Natural Gas | 7390 (88.42%) | 968 (11.58%) | 8358 |
| Aqueduct | 2623 (94.6%) | 150 (5.4%) | 2773 |

Figure 7 shows the behavior of the entire corpus by month, and it is visualized that approximately 60% of all tagged tweets correspond to a negative sentiment, about 25% correspond to a positive sentiment, and the remaining correspond to a neutral sentiment.

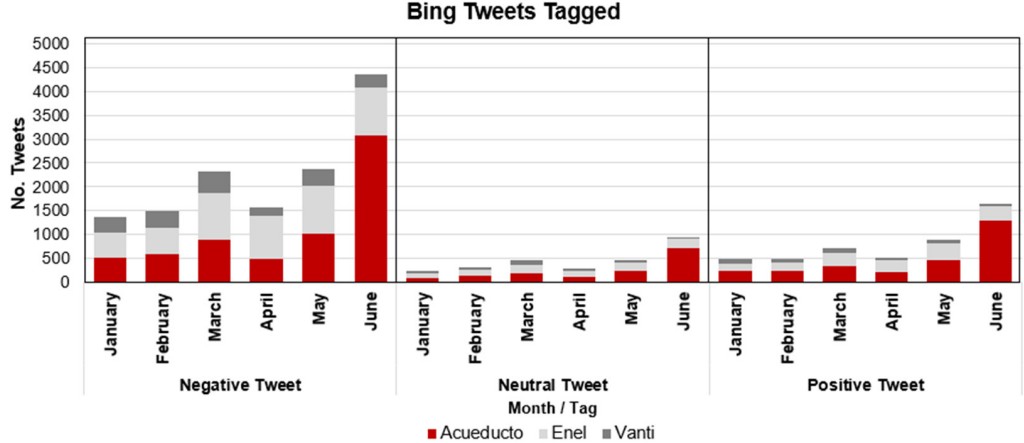

**Figure 7.** Monthly temporal analysis of the distribution of tweet tagging through BING for the Acueducto.

In addition, for the Aqueduct, the month of June had the highest number of negative messages (47% of the total negative tweets), influenced by problems with service outages. For Enel, it can be observed that on average, 67.7% of the tweets are labeled negative, 20.11% are labeled neutral, and the remaining are labeled positive. Finally, for Vanti, it can be observed that on average, 72% of the tweets received monthly correspond to negative tweets and 22% correspond to neutral tweets. In general, for the three companies, the proportion of tags is maintained over time.

According to the results shown in Figure 8, for the three companies, the highest number of negative tweets is obtained on Thursdays and Fridays (on average 65%) compared to the other days. These 2 days represent a negative temporal block that should be considered by quality service analysts for the three companies.

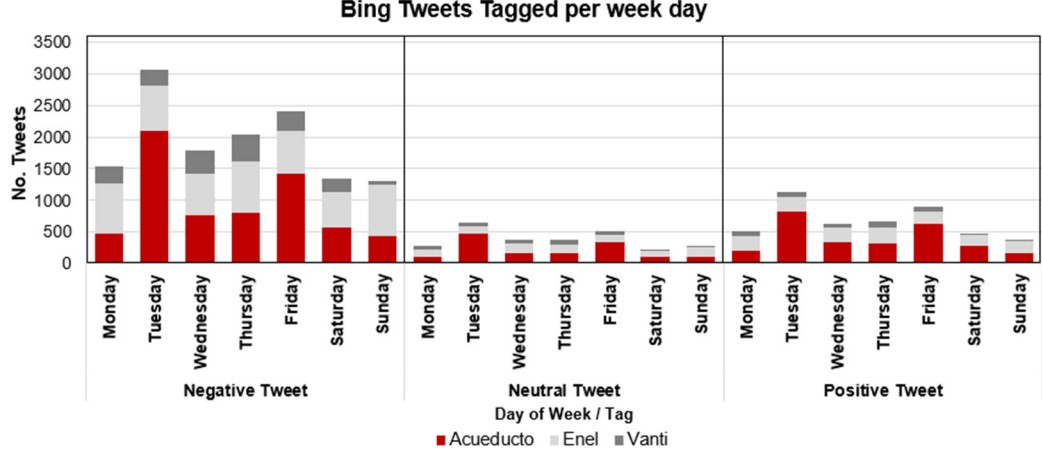

**Figure 8.** Temporal analysis by day of the week of the distribution of tweet tagging by using BING for the three companies.

4.2.2. Evolution of Feelings with AFINN

The AFINN dictionary allows quantifying the intensity of each word associated with its respective sentiment on a scale from −1 to 1, where the daily average of the word scores is taken to get the time series that were analyzed with the "Loess" method. Loess was applied due to the following reasons:

- It is characterized by the fact that the larger the interval of the analyzed data, the smoother the resulting curve, and the better performance will be obtained by using a considerable volume of data.
- The method eliminates noise, and it helps to observe characteristics and trends in a friendly and easy manner.

The results of Figure 9 correspond to the Acueducto. There is high variability until mid-April, but then there is a kind of stabilization around 0 for the rest of the time. Regarding the applied smoothing, the values are close to 0 with a slight tendency to be positive and tend to decrease at the end of the period. The grey shade area indicates the variation of temporal series smoothed and most of points are contained here because the values is near to 0.

In the case of Enel (Figure 10), on most days there are scores above 0, so it can be inferred that there is a tendency to transmit slightly positive messages. The same pattern could be seen with the smoothed series, according to AFINN (the graph shows that most of the points in the graph are contained in the smoothed interval and maintain a stable trend over time).

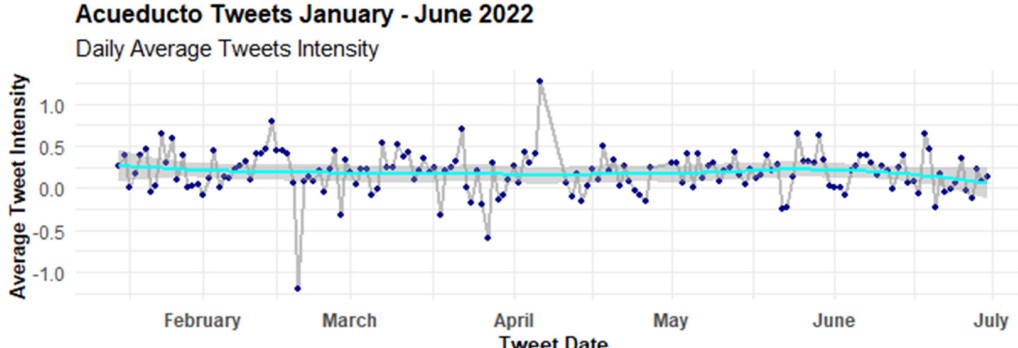

**Figure 9.** Temporal analysis of sentiment intensity through AFINN for Acueducto.

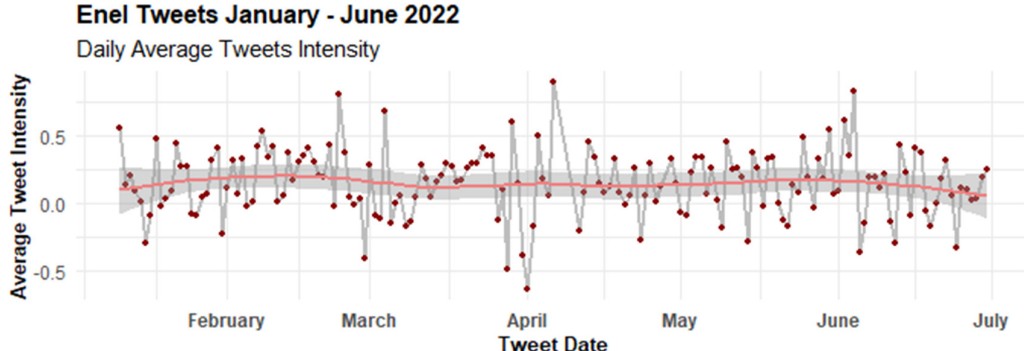

**Figure 10.** Temporal analysis of sentiment intensity through AFINN for Enel.

Finally, for Vanti, it can be observed in Figure 11 that the results present low variability with a couple of outliers that do not affect the smoothing of the data. The smoothed curve has a reduced variation interval, so it can be concluded that the data are homogeneous and behave stably over time.

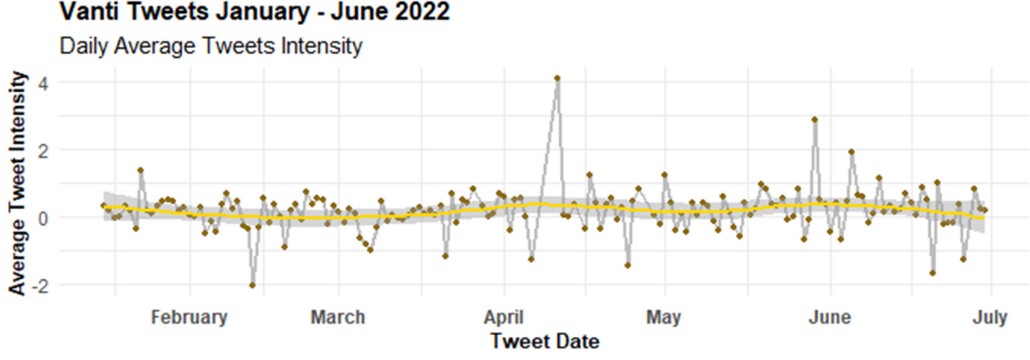

**Figure 11.** Temporal analysis of sentiment intensity through AFINN for Vanti.

### 4.2.3. Evolution of Sentiments with the NRC

To analyze the evolution of sentiments through the NRC dictionary, the number of words assigned to each tag was totalized and multiplied as many times as their frequency of occurrence in each tweet; this was executed to obtain the percentage share of the corpus in each category.

For the Acueducto, the results shown in Figure 12 indicate that users convey feelings associated with Trust, Sadness, and Fear, which were dominant in the period analyzed during this research. Positive feelings such as Surprise and Joy occupy the last places in the labeling in all months. An important finding to highlight is that the Anticipation sentiment

is dominant in the months of April and May, so it can be inferred that user messages anticipate ambiguous responses from the company instead of receiving a direct response.

**Figure 12.** Temporal analysis of sentiments through the NRC to Acueducto.

In the case of Enel, Figure 13 shows that Trust, Sadness, and Fear predominate. Here again, the feeling of Anticipation has the same meaning as in Acueducto.

**Figure 13.** Temporal analysis of sentiments through the NRC to Enel.

For Vanti, in Figure 14, the results of the evolution of feelings over time (monthly) can be observed. Four categories predominate: Anticipation, Trust, Fear, and Sadness.

### 4.3. Topic Modeling and Trends

#### 4.3.1. The Optimal Number of Topics

The identification of the main topics that customers mention in the social network is performed by the "topicmodels" library of the R software [21] under the LDA algorithm. However, it is imperative to previously define the optimal number of topics for each company. For this purpose, the "FindTopicNumbers" function of the "Ldatuning" package of the same program was implemented.

The result of the application corresponds to the graphical representation of four metrics: "Griffiths2004", "Deveaud2014", "CaoJuan2009", and "Arun2010". The objective is to identify the number of topics where the first two metrics present a maximum value and the other two a minimum value. The result for Acueducto, Enel, and Vanti is presented in Figure 15.

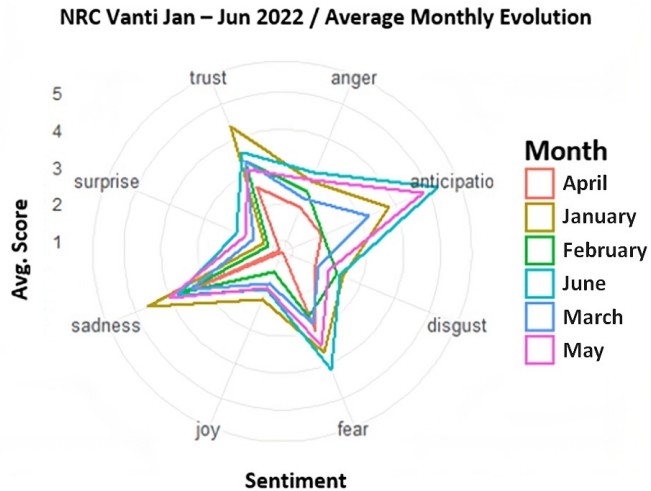

**Figure 14.** Temporal analysis of sentiment through the NRC to Vanti.

From Figure 15, it is determined that the optimal number of topics for all the companies ranges between five and six; however, when applying the LDA algorithm with K = 6, it is found that some words are present in more than one topic, that is, they are repeated in different lists of grouped words. For this reason, the final choice for K was K = 5.

4.3.2. Main Topics

The LDA algorithm uses the top 10 terms with the highest probability of occurrence for each of the five predefined topics (K = 5) and thus assigns a name to each group where the main topics that the customer talks about on Twitter are identified (Table 5).

**Table 5.** Name assignment by topic and company.

| Topic | Acueducto | Enel | Vanti |
|:---:|:---:|:---:|:---:|
| 1 | Water truck request | Services outages | Billing |
| 2 | Services outages | Case resolution and information request | Services outages |
| 3 | Infraestructure damage | Billing | On-site review |
| 4 | Works on public roads | Public lighting | Poor services—PQRS * |
| 5 | Case resolution and information request | Damage household appliances | Case resolution and information request |

* P (Questions), Q (Complaint), R (Claims), and S (Request).

To evaluate the evolution of VOC over time, a dominant topic must be assigned to each tweet of each user of the three companies. To perform this massive assignment, the following is required: the list of words for each topic with their respective probability of appearance, and the words in each tweet. Subsequently, a weighting is performed between the number of words in each message and the probability of occurrence of each term in each topic to obtain the prevalence of a single topic in each tweet (the dominant topic) based on the probability of occurrence of the words used in each topic.

The distribution of the topics at the general level for each company can be seen in Figure 16, which was constructed using Power BI [22]. It is emphasized that more than 90% of all tweets were assigned to one topic, and the remaining ones were assigned to the "other" category. From a general point of view, each company has a dominant topic in which more than 25% of the communications are classified. Thus, for the Acueducto, it corresponds to "Water Truck Request", for Enel, it is identified with "Service Outages" and for Vanti, it corresponds to "Case Resolution and Information Request".

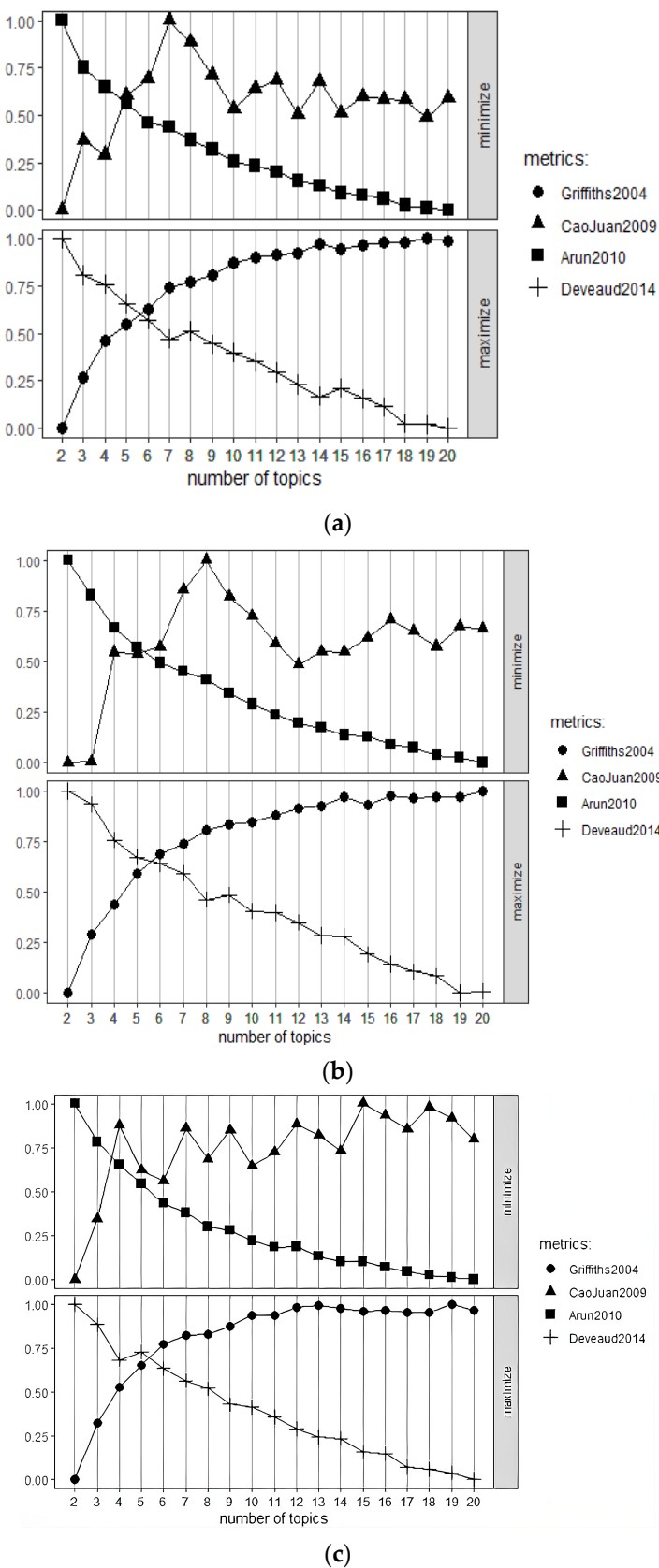

**Figure 15.** The optimal number of topics for: (**a**) Acueducto de Bogotá, (**b**) Enel, and (**c**) Vanti.

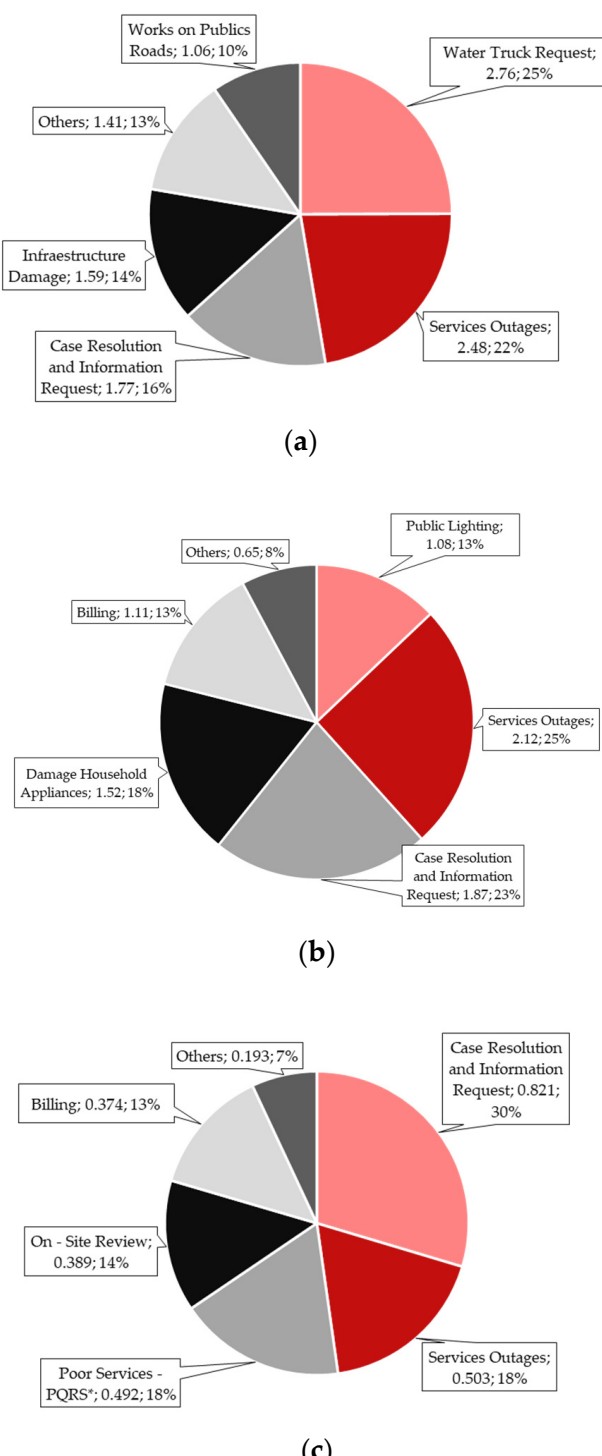

**Figure 16.** Distribution of topics (thousands of tweets) (**a**) Acueducto, (**b**) Enel, and (**c**) Vanti. * P (Questions), Q (Complaint), R (Claims), and S (Request).

To deepen the understanding of the topics, a word cloud is constructed for each topic in each company; this result is presented in Appendix B. Through these visualizations, it is inferred that the terms involved in the tweets classified in each topic correspond to the assigned title and therefore present a direct relationship.

### 4.3.3. Temporal Evolution of Topics

After the identification of the dominant topic of each tweet, it is possible to perform a temporal analysis focused on the previously exposed topics, specifically in the days of the week and during the six months of the study, to identify trends, critical days, and cyclicity, among others. The visualizations presented are also built in Power BI [22].

Figure 17 shows the behavior of the general contactability of the topics of each company. The days with the highest traffic for the Acueducto are Thursdays and Fridays, in which the dominant topic is trending ("Water Truck Request") as well as "Service Outages". For Enel, uniform behavior is evidenced throughout the week (except on Saturdays), especially in the tweets related to "Case Resolution" and "Service Outages" (dominant topics, with high traffic on Sundays). Finally, for Vanti, weekends are days with very little contactability; however, from Monday to Friday, the dominant topic is seen to be "Case Resolution" which presents peaks on Tuesdays, as well as "On-site Review".

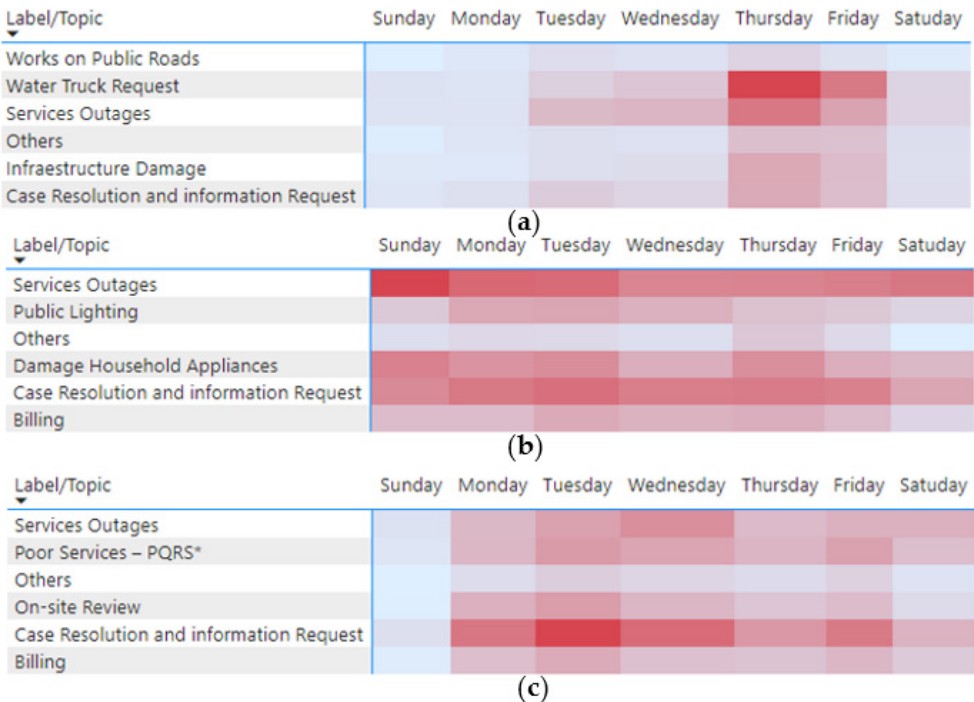

**Figure 17.** Heatmaps group by day of week and topic (**a**) Acueducto, (**b**) Enel, and (**c**) Vanti. * P (Questions), Q (Complaint), R (Claims), and S (Request).

On the other hand, Figure 18 represents the monthly evolution of the topics for each company. For example, for Acueducto, it is evident that in June there is a peak of more than 2000 tweets since on the 23rd and 24th there is a massive failure of service exceeding 24 h, and for more than one neighborhood of Bogota, those customers jointly report the situation. As a result, the topics "Water Truck Request" and "Service Outages" had an increase of about 50%.

For Enel, it is displayed in Figure 19 that there has been an increase in the use of the channel during the last 3 months. The topics, in general, have stable behavior (in comparison with the other two companies), especially for "Service Outages", so it is inferred that the channel is used to report and know the status of these failures.

Finally, for the Vanti company, Figure 20 shows a slight growth in terms of its dominant topic ("Case Resolution and information Request") until March; from then on, there is a gradual decrease.

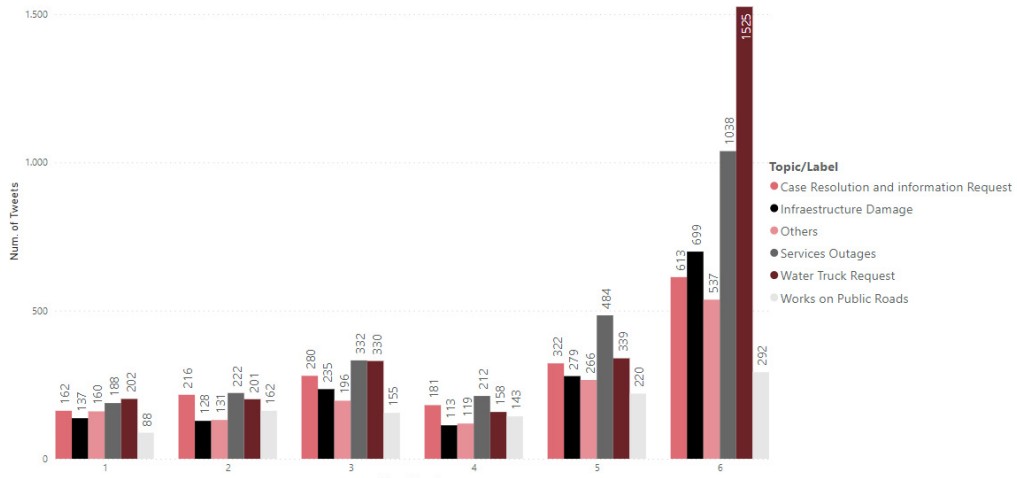

**Figure 18.** Monthly evolution of topics for the Acueducto.

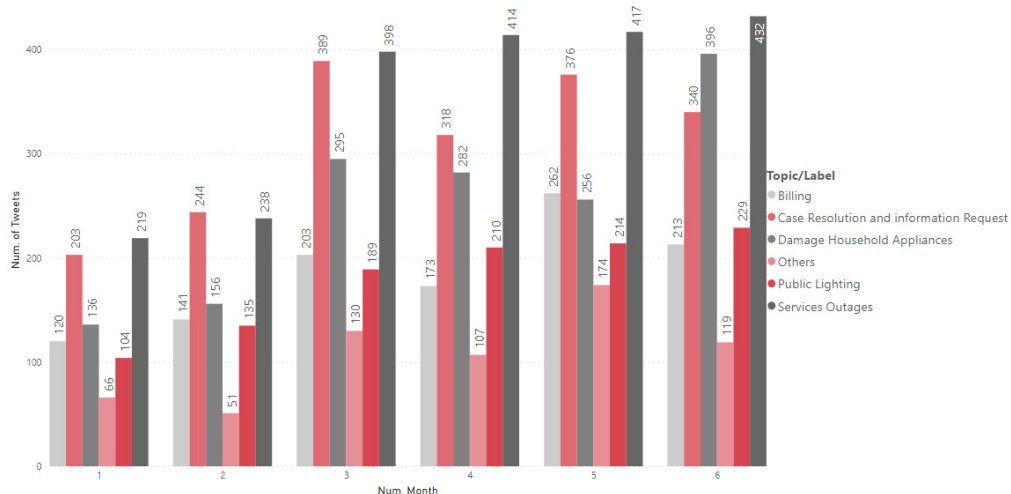

**Figure 19.** Monthly evolution of topics for Enel.

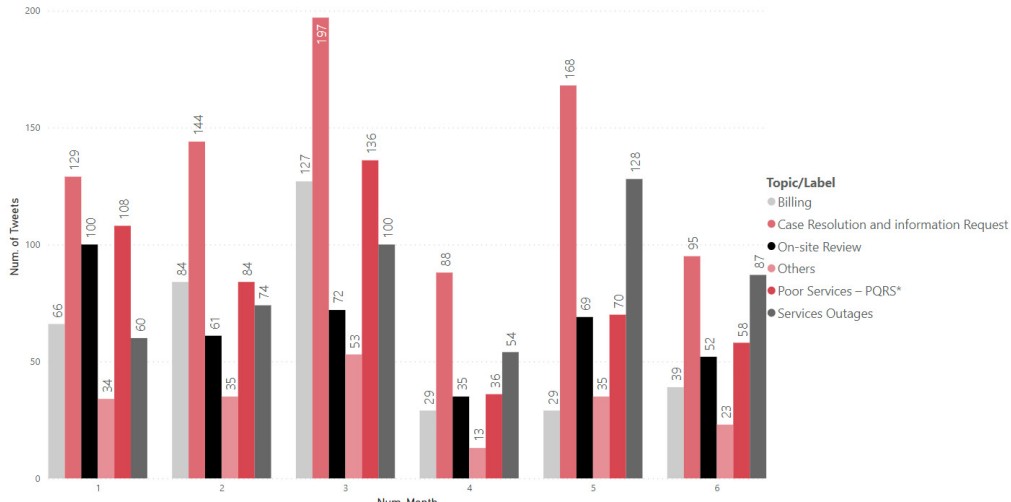

**Figure 20.** Monthly evolution of topics for Vanti.

### 4.3.4. Critical Topics

In Section 4.3.2, the main topics mentioned by the customers of the three companies are presented. However, it is not yet identified which topics generate the most pain and how to connect these problems to specific areas or departments in the companies to improve customer perception. To accomplish this, sentiment analysis is performed on the tweets on each topic, and heatmaps are made to identify in the six months of the study which areas are the most critical.

As explained in Section 3, the Bing dictionary is used to assign a polarity to each tweet since this lexicon has a higher percentage of tweet classification than AFINN. On the other hand, at a general level, it can be identified that about 56% of the communications are categorized as negative tweets and 21% as positive tweets.

Under these premises, it can proceed to cross-check the positive, negative, and neutral tweets corresponding to each of the topics. The results are presented in Appendix **??**. For Enel, "Service Outages" are the biggest customer pain during the evaluation period. This topic has had a sharp increase since March, reaching about 300 negative tweets per month. "Damage Household Appliances" and "Case Resolution and information Request" are the other two ailments that generate an impact on the perception of service.

Regarding the Acueducto of Bogota, the "Water truck request" with more than 1162 negative tweets, followed by "Service Outages" which starts with 150 negative communications and ends with more than 300, are the most common problems and claims reported by the users.

Finally, for Vanti, there is an increase during the first months of the study; in May, there is an inflection point where some topics show a decrease in negative tweets. "Case Resolution and information Request" is the central pain. "On-site Review" is another topic that shows a downward trend in the first four months of the year but then reaches a considerable percentage of negative tweets.

### 4.4. Metrics

Finally, some metrics are proposed as a complement to monitoring and analyzing users' feedback on Twitter. These metrics are a kind of KPI that can be easily calculated in different blocks of time to report the VOC in social networks and its relationships with the quality and perception of the service studied within the three Colombian companies. These suggested KPIs can be incorporated into visualization tools as dashboards to support decision-making.

These metrics are described in detail in Appendix C.

## 5. Discussion

This study exemplifies how social networks are evolving into valuable sources of unstructured data. When appropriately modeled, organizations can extract tactical and strategic information for decision-making without requiring significant investments in development and technology.

For example, text mining has gained significant importance in generating value and comprehending public sentiment regarding specific subjects. In this instance, the study concentrates on identifying the primary sources of dissatisfaction among customers in Bogota's public services sector, uncovering that "Service outages" stand out as the main concern for the users.

The definition of methods for sentiment analysis under the dictionary perspective was conducted to increase accuracy and obtain more robust results aligned with the reality being studied. As a result, the BING and AFINN dictionaries achieved better performance, with 70.9% and 63.6%, respectively. Furthermore, for the NRC dictionary, better results were obtained using pre-defined functions in R specifically designed for this purpose.

An important contribution of this research lies in the analysis of the temporal evolution of sentiment and the recognition of how it changes over time, enabling the identification of patterns, trends, and outliers. Finally, the results allow for the identification of key problems

for the improvement of areas across various service fronts for each company, as well as specific issues to be addressed to enhance processes and improve the user experience with the services provided.

The application of topic modeling through the LDA algorithm provided the determination of the optimal number of topics (k). However, the literature consulted lacks detailed guidance on selecting this number, which highlights it as an important aspect of this research. The graphical method provides a useful approximation to the dataset, but it necessitates subsequent analysis of each group of words to evaluate their coherence and relevance to the word's aggrupation. Multiple iterations were performed to evaluate and enhance the results, involving numbers of k close to the identified optimal value.

The analysis of topics revealed common themes mentioned by customers of the three companies on Twitter, primarily related to "Service Outages" and "Case Resolution and Information Request". These topics accounted for 20% and 23% of the entire corpus, with approximately 4.47 thousand and 5.1 thousand tweets, respectively. The "Others" category was assigned to only 2249 tweets, less than 10% of the downloaded messages, indicating a good approximation by the LDA algorithm. It is recommended to conduct a more in-depth analysis to identify subtopics within critical topics by extending the study period to more than six months and incorporating an additional layer of topic modeling.

When examining the monthly behavior of topics for the companies, a consistent trend in critical topics becomes apparent, as the topics with the highest number of associated tweets remain the same month after month. For the aqueduct company, "Water Truck Request" and "Service Outages" stand out, while for Enel, failures or claims for "Service Outages" represent the most frequent problem that needs to be solved. For Vanti, "Case Resolution and Information Request" takes precedence. Therefore, it is recommended to conduct a more comprehensive analysis to identify subtopics within each critical topic, extending the study period beyond six months and employing an additional layer of topic modeling.

Heatmaps used to identify critical topics for each company proved to be effective visual tools for promptly recognizing customer concerns. The analysis indicates that the overall perception is not favorable, as negative tweets have increased. "Service Outages" emerged as the critical topic for Acueducto and Enel, while Vanti experienced the most significant issues related to "Case Resolution and Information Request". However, Vanti exhibited a downward trend in negative tweets, suggesting that its perception could be improved. In summary, Enel has problems with power supply, especially during the rainy season. Something similar happens with Acueducto, but in this case, the problem gets worse because it seems to be that responses related to water trucks present delays when water service is not provided. Vanti appears to have fewer problems in terms of service supplies; its problems refer to topics associated with administrative failures (case resolution and on-site reviews). These insights not only report the voice of the customer but can also be used by the local authorities to control and supervise these companies.

In conclusion, this research demonstrates the possibility of developing a metrics model for improved monitoring by classifying and categorizing tweets based on topics and dictionaries. By leveraging the voice of the customer as the primary input, this approach provides valuable insights for monitoring the performance of channels such as Twitter, enabling the implementation of tactical actions aligned with customer-centric strategies.

Furthermore, this research proves that it is possible to transform qualitative and unstructured data into information by assigning quantitative values. This conversion and assignment process was facilitated by tools like R [21] and Power BI [22], which were used to construct visualizations suitable for managerial decision-making based on data-driven approaches. This research also follows a methodology for extracting valuable information from digital channels, particularly Twitter, and leveraging it as strategic assets of high relevance for commercial areas, marketing, and business intelligence.

## 6. Future Works

A list is presented for future research according to the results of the research:

- Language: enriching the dictionaries used for sentiment analysis with terms from the Latin-Spanish language will allow wider coverage of the corpus, improving the score assigned to each tweet, the tagging per word, and the homologation with the user survey scales;
- Sources of information: considering other sources of information provided by each of the companies can increase the size of the corpus, contributing to the generation of an automated model of experience management where other channels of attention are involved to identify issues of general interest as well as new knowledge that not only social networks can provide;
- Smart Cities: the use of text mining in social networks can be of potential use for projects related to smart cities since, by labeling this data, it is possible to identify social issues related to the provision of basic services and generate impact projects that improve the quality of life of users;
- Machine Learning: Considering the automation of text mining models and techniques at an unsupervised level will facilitate the creation of self-learning algorithms that can interpret according to context the message that each user transmits to each organization and create personalized attention flows. Likewise, this can be used to enrich the dictionaries used for sentiment analysis, train the algorithms to refine and improve the results of topic modeling;
- Identify irony, sarcasm, and mockery within texts: an important advance that can be made in the field of text mining to identify sentences, tweets, and messages in general is determining the irony with which the sender expresses himself or herself to obtain an objective context of the intentions of the message and what he/she wants to convey to the receiver.

**Author Contributions:** Conceptualization, D.C., V.E.O., C.E.G. and J.G.J.; methodology, D.C., V.E.O., C.E.G. and J.G.J.; validation, D.C., V.E.O., C.E.G. and J.G.J.; formal analysis, D.C., V.E.O., C.E.G. and J.G.J.; investigation, D.C., V.E.O., C.E.G. and J.G.J.; writing—original draft preparation, D.C., V.E.O., C.E.G. and J.G.J.; writing—review and editing, D.C., V.E.O., C.E.G. and J.G.J.; visualization, D.C., V.E.O., C.E.G. and J.G.J. All authors have read and agreed to the published version of the manuscript.

**Funding:** This research received no external funding.

**Institutional Review Board Statement:** Not applicable.

**Informed Consent Statement:** Not applicable.

**Data Availability Statement:** Data will be made available upon request.

**Conflicts of Interest:** The authors declare no conflict of interest.

## Appendix A

**Table A1.** Word clouds by topic for Enel (this exercise was performed for the other two companies).

| Company | N° | Topic | Word Cloud | Analysis |
|---|---|---|---|---|
| Enel | 1 | Service Outages |  | It should be noted that for this topic the words "Cundinamarca", "Calera", "municipio", and "vereda" appear, which indicates that this channel is also used at the rural level as a means of reporting failures, especially in the municipality of La Calera. Terms such as "failures", "again", "maintenance", and "service" appear, which means that there are service interruptions. |
| | 2 | Case Resolution and Information Request |  | For topic two, words such as "answer", "internal", "need", "message", and "response" are evidenced, which means that customers are waiting for their answers, that they have sent the data of their "cases" and that they may require an effective response to what they indicate through the channel. |
| | 3 | Billing |  | For the topic "Billing", there is the appearance of words such as "pay", "receipt", "arrives", and "invoice", which revolve around the topic of payments and invoicing. It is evident that customers ask about the payment of their "receipts" through this network and that they may be inconvenienced since there are terms of annoyance such as "angry". |

**Table A1.** *Cont.*

| Company | N° | Topic | Word Cloud | Analysis |
|---|---|---|---|---|
| | 4 | Public Lighting | urgente luz localidad electrico norte senores sector mas suba barrio energia sur estan publico zona calle ayuda parque falla poste luminarias servicio alumbrado | This topic is related to power outages but directly relates to "poles" and "luminaires". Terms such as "park", "urgent", "zone", and "public" suggest that customers report public lighting failures on the social network and that, in some cases, they may be "urgent". Terms such as "neighborhood" and "locality" show that the address of the new development is shared. |
| | 5 | Damage Household Appliances | pueden barrio llevamos mas senores manana sector cortes menos cada mal van danc responde dos servicio semana luz danos veces va electrodomesticos energia | As can be seen, this topic is closely related to topic one, since words such as "outages", "damages", and "services", among others, appear. This indicates that, given the micro-cuts and interruptions in the electric power supply, the customers' "electrical appliances" burn out, so the user chooses to use this channel as a means of resolution. |

## Appendix B

This section presents the heatmaps corresponding to the evolution of the five topics for each company and the categorization into positive, negative, and neutral of the tweets of each of the messages included in each topic (this exercise was also performed for Acueducto and Vanti). The red color characterizes tweet categories with a higher message count, yellow indicates a moderate amount, and green represents categories with fewer messages.

| Topic/Label | Number of Month | | | | | | |
|---|---|---|---|---|---|---|---|
| Bing´s Categorization | 1 | 2 | 3 | 4 | 5 | 6 | Total |
| Services Outages | 219 | 238 | 398 | 414 | 417 | 432 | 2118 |
| Negative Tweet | 149 | 155 | 278 | 282 | 311 | 309 | 1484 |
| Neutral Tweet | 36 | 48 | 64 | 70 | 51 | 70 | 339 |
| Positive Tweet | 34 | 34 | 53 | 61 | 55 | 53 | 290 |
| Untagged Tweet | 0 | 1 | 3 | 1 | 0 | 0 | 5 |
| Case Resolution and Information Request | 203 | 244 | 393 | 319 | 376 | 340 | 1875 |
| Negative Tweet | 123 | 139 | 246 | 187 | 230 | 228 | 1153 |
| Neutral Tweet | 34 | 53 | 83 | 75 | 75 | 55 | 375 |
| Positive Tweet | 46 | 52 | 64 | 57 | 71 | 57 | 347 |
| Damage Household Appliances | 136 | 156 | 295 | 284 | 256 | 397 | 1524 |
| Negative Tweet | 82 | 97 | 185 | 167 | 157 | 205 | 893 |
| Neutral Tweet | 36 | 42 | 70 | 80 | 64 | 138 | 430 |
| Positive Tweet | 18 | 16 | 39 | 37 | 35 | 52 | 197 |
| Untagged Tweet | 0 | 1 | 1 | 0 | 0 | 2 | 4 |
| Billing | 120 | 141 | 204 | 173 | 262 | 213 | 1113 |
| Negative Tweet | 76 | 80 | 121 | 96 | 134 | 118 | 625 |
| Neutral Tweet | 27 | 38 | 45 | 51 | 85 | 51 | 297 |
| Positive Tweet | 17 | 23 | 38 | 26 | 39 | 44 | 187 |
| Untagged Tweet | 0 | 0 | 0 | 0 | 4 | 0 | 4 |
| Public Lighting | 104 | 135 | 189 | 210 | 214 | 229 | 1081 |
| Negative Tweet | 61 | 72 | 99 | 118 | 105 | 109 | 564 |
| Neutral Tweet | 27 | 38 | 50 | 56 | 66 | 60 | 297 |
| Positive Tweet | 16 | 25 | 38 | 36 | 43 | 58 | 216 |
| Untagged Tweet | 0 | 0 | 2 | 0 | 0 | 2 | 4 |
| Others | 66 | 51 | 130 | 107 | 174 | 119 | 647 |
| Negative Tweet | 17 | 26 | 47 | 31 | 41 | 28 | 190 |
| Neutral Tweet | 22 | 10 | 34 | 33 | 43 | 26 | 168 |
| Positive Tweet | 18 | 8 | 29 | 22 | 59 | 31 | 167 |
| Untagged Tweet | 9 | 7 | 20 | 21 | 31 | 34 | 122 |
| Total | 848 | 965 | 1609 | 1507 | 1699 | 1730 | 8358 |

**Figure A1.** Heatmap by topics—Enel.

## Appendix C

**Table A2.** Tracking metrics.

| Metrics | Objective | Temporality |
|---|---|---|
| Number of tweets per hour by topic | Identify topics that may be trending within each company in advance to take corrective actions. | Per hour |
| Number of negative tweets on the critical topic by the company | Based on the identification of the topics with the highest number of negative tweets (categorized through the Bing dictionary), it is proposed to perform daily monitoring of the number of communications that are categorized within this group. As in previous cases, we can identify actions that have generated a decrease in this indicator and that can be maintained over time to generate a continuous improvement. | Daily |
| Tweet polarity score (tps) | Determine the number of positive and negative tweets according to the Bing dictionary classification. | Per hour |

**Table A2.** *Cont.*

| Metrics | Objective | Temporality |
|---|---|---|
| Pn Tweet polarity score (pn Tweet) | Determine the ratio between the total number of positive and negative tweets, always aiming to be greater than or equal to one. It makes it possible to quickly track the general perception of the user. | Per hour |
| User polarity score (up) | Identify the category (positive, negative, neutral, and n/a) of users from the tagging of their tweets, focusing on identifying which users have a higher amount of positive or negative tweets. It is based on the Bing dictionary | Per hour |
| Pn user polarity score (pn user) | Determine the ratio between users categorized as positive and negative according to the Bing dictionary. Like the previous metric, it shows a ratio, but this time between the number of positive and negative users, so it should aim to be at least equal to one. | Per hour |

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
