# Peer review of "Monitoring the Quality and Perception of Service in Colombian Public Service Companies with Twitter and Descriptive Temporal Analysis"

_applsci, doi:10.3390/app131810338_

Round 1
Reviewer 1 Report
1. Authors fails to raise research gap in section of Introduction, so the research problems are not clearly demonstrated. Therefore, the innovation of this work is not obvious.
2. The second section should be modified into Literature Review, and should be well organized to present the streams of related research.
3. The methodological basis of this study is very weak.
4. It is recommended that the author add a research framework for this article, and elaborate on the research ideas, research steps, research methods, etc., to help readers better understand this article.
5. The text in the picture is not clear, so the authors need to modify the picture format.
6. Authors should supplement the part of conclusion to include the main findings.
7. References are wrong cited and citations format should be appropriate modifies.
English writing is poor and requires comprehensive revision.
Reviewer 2 Report
The objective of this case study appears to be monitoring the quality and perception of service in Colombian public service companies through the analysis of Twitter data. This study aims to gain insights into providing better customer services to public utility companies and understanding well-known text mining techniques and metrics. The findings of this paper are interesting, although the research focuses only on three companies and utilizes classic techniques. The manuscript requires improvements before it can be considered for publication. Here are my comments for the authors:
Abstract:
1. I suggest that the authors clarify the research objective in the abstract.
2. "Additionally, this study presents the homologation process for three out of the five pillars on which the CIER survey is based."
-This sentence lacks clarity and requires more context and explanation regarding the "homologation" process of the pillars of the CIER survey. Consider removing it.
Introduction:
3. The introduction lacks a clear and structured flow and is unnecessarily long. I recommend removing the last two paragraphs. Pay attention to the topic sentence as it signifies the content of each particular paragraph.
4. While the introduction mentions the objective of identifying critical areas for improvement, it could be stated more explicitly and clearly. I recommend placing it at the beginning of a paragraph to ensure readers can understand it clearly.
Background and previous research:
5. "Another contribution to the present research that supports the inclusion of customer centricity in company decisions is found in [7]."
-Does "present research" refer to the authors' research? Why is another contribution mentioned? This sentence is unclear. Perhaps the authors meant to refer to previous research [7].
6. The authors use the word 'another' frequently in this section, which reduces readability.
7. The literature review lacks a clear structure and flow of ideas. The paragraphs jump from one study to another without providing proper transitions or connecting sentences. It would be helpful to group related studies together and present them in a logical order.
Materials and Methods:
8. It would be helpful to divide this section into subsections based on different stages or processes of the methodology (e.g., data collection, data preprocessing, sentiment analysis).
9. The table provided in the methodology section (Table 1) shows the results of the evaluation of sentiment analysis methods, but the actual results are not presented in the text. It is important to include the results in the text or provide a clear reference to the table.
Results:
10. Figure 3, 4, and 6 could be combined into one figure consisting of subplots. This would allow for easier comparison.
11. Figures 7, 8, and 9 could be combined.
12. Figures 10, 11, and 12 could be combined.
13. Figures 13, 14, and 15 could be combined.
14. The fonts shown in Figure 17 seem to be too small."
There are too many language-related issues in this paper. It is important for the authors to address these issues before submitting the paper to an academic journal.
Reviewer 3 Report
Remarks, comments and questions to be answered.
What are the research questions in this paper? What is the main reason for the presented investigation?
The cited references are relevant to the research but limited. The authors should show previous research in a broader context.
The research design is appropriate and acceptable.
The fundamental question is, what are the practical implications of this study? It has to be emphasised in conclusions.
Round 2
Reviewer 1 Report
Thanks authors for their great effort on modifying the manuscript and the revisied vision has almost responded my concerns. However, I think the citation format should be checked again after referring the previous published work.
Reviewer 2 Report
Thank you for addressing the concerns I raised in the earlier review. The revised version is a substantial improvement compared to the previous one.
Although the language used is acceptable, there is room for improvement in the writing to enhance readability and flow more naturally. Additionally, this paper requires some typographic corrections. For instance, line 586 contains an incomplete sentence.
